# Study on Associating Emotions in Verbal Reactions to Facial Expressions in Dementia

**DOI:** 10.3390/healthcare10061022

**Published:** 2022-06-01

**Authors:** SungHo Hwang, JiWon Hwang, HyeonCheol Jeong

**Affiliations:** 1College of Nursing, Sahmyook University, Seoul 01795, Korea; sungho@xxhwangxx.net; 2College of Nursing, Kyungdong University, Wonju 24695, Korea; chiwon0909@kduniv.ac.kr

**Keywords:** elderly with dementia, emotion of verbal expression, emotion of facial expression

## Abstract

The purpose of this study was to provide basic data on cognitive therapy and to improve social support programs for the elderly with dementia by identifying the difficulties they experienced in emotional communication by identifying how they recognized emotions in verbal reactions to facial expressions using Ekman’s photographs of facial expressions and comparing their responses with the general elderly population. There were 141 participants in this study. Data collection was conducted from 3 April 2019 to 30 June 2019 in Seoul, in the Gyeonggi-do and Gangwon-do provinces of South Korea. This study performed descriptive research in which subjects made participative decisions with their guardian through recruitment. The tools used in this study included a general characteristic questionnaire and the Ekman 6 facial expressions photographs tool, which underwent intensive validity studies. The collected data were analyzed using the R version 3.5.1 statistic computing platform. The ability of the elderly with dementia to associate verbal expressions with facial expressions differed from that of the general elderly population. The rates of correct associations of verbal expressions to facial expressions were similar across dementia grades. There was a significant difference in the proportion of correct associations between positive and negative emotions in the elderly with dementia compared to the general elderly population. In the elderly with dementia, pictures showing fear, anger, and disgust had higher wrong rates of incorrect answers than correct answers. The average score of elderly with dementia in associating verbal expressions with six facial expressions was 2.69, which was even lower when they were asked to associate verbal expressions with pictures showing facial expressions of anger or disgust. This study shows that elderly persons with dementia have difficulties identifying two negative emotions (anger, disgust) and find it much easier to identify a positive emotion of happiness represented by a smiling face. Since the ability of the elderly with dementia to interpret verbal expressions to facial expressions was different from that of the general elderly population, careful attention and consideration are needed to support and communicate emotions to the elderly with dementia.

## 1. Introduction

### 1.1. Need for Study

According to a report from the World Health Organization (WHO), about 50 million people have dementia. All dementia patients and their family members live with physical, psychological, and social burdens. WHO also points out that around 1.1% of the global GDP (gross domestic product) in 2015 was spent on serious social problems [1,2,3,4]. According to a report by the Korean Dementia Observatory 2018, there are 705,473 estimated dementia patients, which corresponds to one out of ten of people who are 65 years old and above. The costs of dementia management in South Korea are estimated to make up 0.8% of the GDP, which is about 14.6 billion [5]. Dementia is recognized as a national problem as well as a family problem. Elderly with dementia experience difficulties in communicating their feelings with other people as compared to the normal elderly due to emotion recognition disorder which comes from the change in cognition due to dementia [6,7,8,9,10,11,12]. Difficulties that occur in relation to empathy and the ability to exchange and understand shared emotions in interpersonal relationships comes from their inability to properly understand others’ emotional states [13,14,15,16,17]. In particular, it is necessary to understand the deficit in emotion recognition in the elderly with dementia.

Verbal expression refers to the communication of one’s beliefs or opinions through speech or in writing [18,19,20,21], and emotions refer to conscious mental reactions such as anger or fear, subjectively experienced as strong feelings usually directed toward a specific object which are typically accompanied by physiological and behavioral changes in the body [22]. While there are several different emotions, Ekman has identified six basic emotions corresponding to facial expressions, including anger, disgust, fear, happiness, sadness, and surprise [23,24,25,26]. Since these facial expressions may be associated with these six emotions, Ekman came up with six facial expression photos which have proven to be valid and have been widely used for studies on emotion recognition. The Ekman facial expressions photos have also been proven to be valid and reliable for Korean people [23,24,25,26]. Although the recognition of emotions such as disgust and fear is somewhat low, Ekman facial expressions are evaluated as useful emotional stimuli in neuroscience studies on emotions, and are utilized as diagnostic tools applicable to the Korean population. Neuropsychological assessment using facial expression stimuli is simple. After an assessor shows the subject photo slides of the facial expressions, the subject selects a word printed on paper which best describes the facial expressions, and these results are scored and used as a diagnostic tool [27,28,29,30,31]. In the elderly with dementia, wherein cognitive disorder is a major symptom, the association between verbal expressions and facial expressions should be understood. In order to understand why it is difficult for the elderly with dementia to communicate, this study aims to demonstrate the ability of the elderly with dementia to associate emotions in verbal expressions with facial expressions using the Ekman facial expression photo tools and comparing their ability to that of the general elderly population. Definitions of dementia have not mentioned emotion recognition definitively. Hence, studies were needed to conduct on emotion recognition of the elderly who experienced dementia and to seek for alternatives to overcome the difficulty of emotion recognition in dementia elderly. This study contributes to the early detection of dementia through the evaluation of recognizing and associating the emotions in verbal expressions to facial expressions [32,33,34,35,36].

### 1.2. Purposes of Study

This study aimed to understand the ability of elderly with dementia to recognize and associate emotions in verbal expressions with facial expressions as compared to the general elderly, and the detailed purposes were as follows. We (1) considered the general characteristics of the elderly with dementia compared to the general elderly population who participated in this study, (2) compared the rate of correct recognition of emotions in the facial expressions of people with dementia with that of general elderly population, (3) compared the rate of correct recognition of emotions in the facial expressions of the elderly with dementia with that of general elderly population, (4) compared the rates of correct recognition of emotions in the facial expressions across different grades of dementia in the elderly, (5) compared the rates of correct recognition of emotions in facial expressions which were positive or negative, (6) determined the degrees of recognition of emotions in facial expressions, and (7) compared the degrees of recognition for facial expressions to the subjects’ scores in associating verbal expressions with facial expressions.

## 2. Methods of Study

### 2.1. Design of Study

This was an explorative research study conducted to understand the inability of the elderly with dementia to recognize and associate emotions in verbal expressions with facial expressions, specifically in those who had features such as memory disorder, aphasia, apraxia, and agnosia.

### 2.2. Subjects of Study

The subjects included patients who were diagnosed with dementia residing in Seoul City, and the Gyeonggi-do province, Gangwon-do provinces. They participated voluntarily after they, along with their guardian, made participative decisions and signed the consent form which followed ethical principles. The sample size was calculated to a total 143 participants with an effect size of 0.3, significance level of 0.05, and power of 0.8 for setting contingency table for chi-square test between the elderly with and without dementia using G*Power 3.0.10 (Heinrich Heine University, Düsseldorf, Germany). In this study, 72 elderly people with dementia and 69 elderly people as control in which subjects who were analyzed for this study were 141 totally. We followed the same method in recruiting the control sample from the general elderly population. The inclusion criteria for each group were as follows: (1) an elderly person with dementia is a person diagnosed with long-term care dementia grades of 3–6 according to the Act on Long-Term Care Insurance for the Aged, while a (2) general elderly person is a person who does not fall under the long-term care grades according to the Act on Long-Term Care Insurance for the Aged.

### 2.3. Ethical Consideration

Consent to participate in the study was included in the questionnaire. The questionnaire could be completed within 30 min (or about 10 min for the general elderly population), considering the participant’s fatigue while answering the questionnaire. We provide the subjects with a small gift for their participation in the questionnaire. Furthermore, we explained that they could stop answering the questionnaire at any time, with no disadvantage on their part. This study was conducted after receiving approval from the institutional research ethics committee in S university.

### 2.4. Data Collection and Analysis Procedure

The study was conducted from 3 April 2019 to 30 June 2019. Questionnaires for the elderly with dementia were collected from centers for dementia in Seoul, Gyeonggi-do, Gangwon-do. For the general elderly group, questionnaires were collected from community welfare centers in Seoul, Gyeonggi-do province, upon approval of the chief of the center. All in all, we collected 72 questionnaires from the group of elderly with, and 69 questionnaires from the general elderly group.

We progressed data collection with care workers who cared their recipients assisted. As for the procedure, we showed six photo slides displaying the Ekman facial expressions to the subjects, and we had the subject select one word printed on paper which best described the facial expression. We confirmed the association between verbal and facial expressions, identified correct answers, and the subjects’ scores were used to find cut-off values and compared among the six expressions from contingency table.

### 2.5. Study Instruments and Data Analysis Methods

Instruments used in this study included a questionnaire which asked about personal identification and general characteristics, as well as photos of the most intense facial expressions for each emotion taken from Ekman’s facial expression photos. The data collected in this study were analyzed using R version 3.5.1 statistical computing platform. The following methods of analysis were used [37]: (1) General characteristics of dementia and general elderly groups underwent frequency and percentage calculation and a χ^2^ test; (2) the percentage of correct associations of verbal expressions with facial expressions was calculated, and these were subjected to a χ^2^ test; (3) the ratio of correct associations of verbal with facial expressions was calculated; (4) correct answers across dementia grades were compared using percentages and average percentages; (5) correct answers between positive and negative emotions were expressed using percentages and underwent a χ^2^ test for frequency; (6) scores of associating verbal expressions with facial expressions were subjected to *t*-test and receiver operating characteristic (ROC) analysis; and (7) logistic regressions between each correct answer and the six correct answer scores was performed, displaying the results as odds ratios.

### 2.6. Limitation of Study

Long-term care grades as dementia in South Korea were 1 to 6. Grades 1 and 2 were approximately severe dementia with any physical impairment. Grades 3 and 4 were approximately moderate dementia with any physical impairment. Grades 5 or 6 were mild dementia [38]. From exclusion criteria which excluded one who could not understand the study procedure, elderly dementia patients in long-term care with grades 1 and 2, or those who had severe or moderate degrees of mental and physical impairment, were excluded from the study and did not answer the questionnaire.

## 3. Results of Study

### 3.1. Characteristics of Study Participants

There were 141 participants in the study, with 72 elderly dementia patients and 69 in the general elderly group. With regards to sex, the elderly dementia group had 45 females (62.5%) and 27 males (37.5%). The general elderly group had 52 females (75.3%) and 17 males (24.6%). There was no significant difference in the sex between the elderly dementia group and the general elderly group (χ^2^ = 2.14, *p* = 0.142). As for age, there were 41 patients aged 80 or above in the elderly dementia group (56.9%), while there were 41 patients aged 70 and above in the general elderly group (59.4%). There was no significant difference in the ages between groups (χ^2^ = 3.14, *p* = 0.075). Long-term care grades of dementia in South Korea were 1 to 6. Grades 1 and 2 were approximately severe dementia with any physical impairment. Grades 3 and 4 were approximately moderate dementia, and grades 5 or 6 were mild dementia. In this study, 21 patients were classified as grades 5–6 (mild) (29.1%) and 51 were classified as grades 3–4 (moderate) (70.8%). Among all the participants, most had an income of less than 1,000,000 won (Table 1).

### 3.2. Correct Associations between Verbal Expressions and Facial Expressions

In totally, 11 (15.2%) of the 72 elderly dementia patients and 10 (14.4%) of the 69 general elderly participants gave the correct answers for the facial expression of fear. As for the anger expression, 33 (45.8%) of 72 elderly dementia patients and 52 (75.3%) of 69 general elderly participants correctly answered. For the sad expression, 41 (49.4%) of 72 elderly dementia patients and 51 (73.9%) general elderly participants gave the correct answer. For the happy expression, 52 (72.2%) elderly patients with dementia and 67 (97.1%) general elderly participants gave the correct answer. For the disgusted expression, 23 (31.9%) elderly dementia patients and 47 (66.6%) general elderly participants gave the correct answer. Lastly, 37 (51.3%) of 72 in the elderly dementia group and 62 (89.8%) of 69 in the general elderly group gave the correct answer for the surprised facial expression (Table 2).

The facial expression that had the highest number of correct answers was happiness (72.2%), while the facial expression with the least number of correct answers was fear (15.2%) in elderly dementia patients. Some of them were unable to answer as well. The facial expressions wherein the number of wrong answers exceeded that of correct answers included fear (81.7%), anger (51.5%), and disgust (65.0%) (Table 3).

Among the general elderly group, the facial expression that had the most correct answers was happiness (97.1%), while fear had the lowest number of correct answers (14.4%). The rates of wrong answers were 75.3% for fear, 15.9% for disgust, and 14.4% for anger (Table 4).

### 3.3. Ratio of Correct Associations between Verbal and Facial Expressions

The ratios of correct associations between verbal expressions and facial expressions may be seen in (Table 5). All the ratios were less than one, except for fear.

### 3.4. Correct Associations between Verbal Expressions to Facial Expressions across Dementia Grades

The rate of correct associations between verbal expressions and facial expressions across dementia grades may be seen in Table 6. Grade 6 had a rate of 35.1% and grade 5 had a rate of 47.2% (mild grade dementia), whereas grade 4 had a rate of 50.7% and grade 3 had a rate of 44.0% (moderate grade dementia).

### 3.5. Correct Associations between Verbal Expressions and Facial Expressions between Positive and Negative Emotions

Fear, anger, sadness, and disgust are considered negative emotions, while happiness and surprise are considered positive emotions. We determined the difference in the rate of correct answers between positive and negative emotions in dementia patients and the general elderly group. In the elderly dementia group, the average percentage of correct answers for positive emotions was 61.7%, while this was 38.0% for negative emotions. In the general elderly group, the average percentage of correct answers for positive emotions was 93.4%, while for negative emotions, it was 57.5% (Table 7).

### 3.6. Scores of Correct Associations between Verbal Expressions and Facial Expressions between Elderly Dementia Patients and the General Elderly Group

The scores of correct associations between verbal expressions and facial expressions ranged from 0 to 6 points. For the elderly dementia patients, the average score was 2.69 points out of a total of 6 points. For the general elderly group, the average score was 4.14 points. The cut-off value between these two groups was 3.5 points (Table 8).

### 3.7. Logistic Regressions of Each Correct Answer versus Correct Answer Scores on All Six Items

Logistic regression analysis for each correct association (0 to 1) between verbal expressions and facial expressions versus the correct answer scores for all six items (0 to 6) was performed. This was done to understand the impact of answering correctly on one item on the total score.

The odds ratio of associating verbal expressions with facial expressions of fear was 2.75 (1.59~5.86) in elderly dementia patients, and this was 6.53 (2.30~26.55) in the general elderly group (Table 9). The even–odd of the association between verbal expressions and facial expressions was 5.22 in elderly dementia patients, while this was 5.80 in the general elderly group (Table 9). The average correct answer scores of 2.69 in elderly dementia patients and 4.14 in the general elderly group did not reach the even–odd (comparison in Table 10) (Figure 1).

The odds ratio of associating verbal expressions with facial expressions of anger was 6.20 (3.00~18.58) in elderly dementia patients and 5.23 (2.41~14.98) in the general elderly group (Table 9). The even–odd of associating verbal expressions with facial expressions of anger was 2.95 in elderly dementia patients and 3.10 in the general elderly group (Table 9). The average score of correct answers was 2.69 in elderly dementia patients, which was lower than the even odd 2.95. Otherwise, the average correct answer score of 4.14 in the general elderly group was higher than the even odd 3.10 (comparison in Table 10) (Figure 2).

The odds ratio of associating verbal expressions with facial expressions of sadness was 3.46 (2.13~6.51) in elderly dementia patients, and was 6.02 (2.67~18.13) in the general elderly group (Table 9). The even–odd of associating verbal expressions with facial expressions of sadness was 2.29 in elderly dementia patients and 3.22 in the general elderly group (Table 9). The average score of correct answers was 2.69 in elderly dementia patients, which was higher than the even–odd 2.29. Furthermore, the average score of correct answers was 4.14 in the general elderly group, which was higher than the even–odd 3.22 (comparison in Table 10) (Figure 3).

The odds ratio of associating verbal expressions with facial expressions of happiness was 3.77 (2.19~7.68) in elderly dementia patients; otherwise, in general elderly, this was not significant (Table 9). The even–odd of associating verbal expressions with facial expressions of happiness was 1.34 in elderly dementia patients and was 0.496 in the general elderly group (Table 9). The average score of correct answers was 2.69 in elderly dementia patients, which was higher than the even–ºodd 1.34. Furthermore, the average score of correct answers was 4.14 in the general elderly group, which was higher than the even–odd 0.496 (comparison in Table 10) (Figure 4).

The odds ratio of associating verbal expressions with facial expressions of disgust was 3.02 (1.88~5.65) in the elderly with dementia and was 13.14 (4.59~58.44) in the general elderly (Table 9). The even–odd of associating verbal expressions with facial expressions of disgust was 3.84 in elderly dementia patients and 3.65 in the general elderly group (Table 9). The average score of correct answers was 2.69 in elderly dementia patients, which was lower than the even–odd 3.84. Otherwise, the average score of correct answers was 4.14 in the general elderly group, which was higher than the even–odd 3.65 (comparison in Table 10) (Figure 5).

Lastly, the odds ratio of associating verbal expressions with facial expressions of surprise was 2.71 (1.79~4.58) in elderly dementia patients and was 4.35 (1.96~15.44) in the general elderly group (Table 9). The even–odd of associating verbal expressions with facial expressions of surprise was 2.63 in elderly dementia patients and 2.00 in the general elderly group (Table 9). The average score of correct answers was 2.69 in elderly dementia patients, which was higher than the even–odd 2.63. Furthermore, the average score of correct answers was 4.14 in the general elderly group, which was higher than the even–odd 2.00 (comparison in Table 10) (Figure 6).

## 4. Discussion

The study used six photos of facial expressions representing different emotions. We determined that the most intense pictures from the six different sets of emotions could be utilized as meaningful instruments.

In this study, subjects chose verbal expression on each of six face pictures. We identified the difference between dementia patients and control in the above condition [40]. Furthermore, the identification of the face pictures caused the difference between the two groups in the average scores on the even–odds.

The rates of correct answers of the elderly dementia patients were ranked as follows, from highest to lowest: happiness (72.2%), sadness (59.4%), surprise (51.3%), anger (45.8%), disgust (31.9%), and fear (15.2%). We suggest that further studies are necessary to explain the ranks of the rates of correct associations of verbal expressions with facial expressions.

There was no clearly observed trend with regard to the rate of correct associations of verbal expressions with facial expressions among grades of dementia. However, there was a clear difference in the correct associations of verbal expressions with facial expressions between dementia patients and the general elderly group. Further studies are necessary to explain why there was a significant difference in the rate of correct answers between the dementia patients and the general elderly group, but insignificant across grades.

In both the dementia patients and the general elderly group, the highest rate of correct answers occurred for the emotion of happiness. This implies that elderly dementia patients and the general elderly group recognize smiling faces and also understand the emotion of happiness. Therefore, it is important for nurses and caregivers to communicate with and support the patients by smiling.

In elderly dementia papers, the facial expressions that elicited the most wrong answers were fear, anger, and disgust. Of these, the facial expressions of anger and disgust scored higher than other facial expressions in terms of their even–odds. The even–odd of anger was 2.95 points while that of disgust was 3.84 points, which were both higher than the average score of 2.69 points. Further studies are necessary to determine if portraits of anger and disgust may be used in simple screening tests for the recognition of facial expression.

## 5. Conclusions and Recommendations

This study shows that elderly persons with dementia have difficulties identifying two negative emotions (anger, disgust) and have much easier identifying a positive emotion of happiness represented by smiling face.

The use of two facial expressions (anger, disgust) as a simple screening test for the recognition of facial expression in dementia patients might be considered in clinical uses and are needed for further studies and more interest of emotion recognition in dementia patients nationally and internationally.

Although the elderly with dementia experienced confusion in recognizing emotions in portraits, their recognition of happiness (smiling face) was high. These imply that elderly with dementia can understand the emotion of happiness and simultaneously recognize smiling faces. Therefore, it may be important for nurses and caregivers to communicate and support their elderly dementia patients by smiling at them.

As a result, our study showed that the ability of elderly dementia patients to recognize and associate emotions in verbal expressions with facial expressions is different from that of the general elderly population. Therefore, more careful attention is needed in communicating with and supporting the emotions of the elderly with dementia, and it is necessary to understand the relative and intrinsic difficulties in communicating the emotions which elderly dementia patients experience [41].

## Figures and Tables

**Figure 1 healthcare-10-01022-f001:**
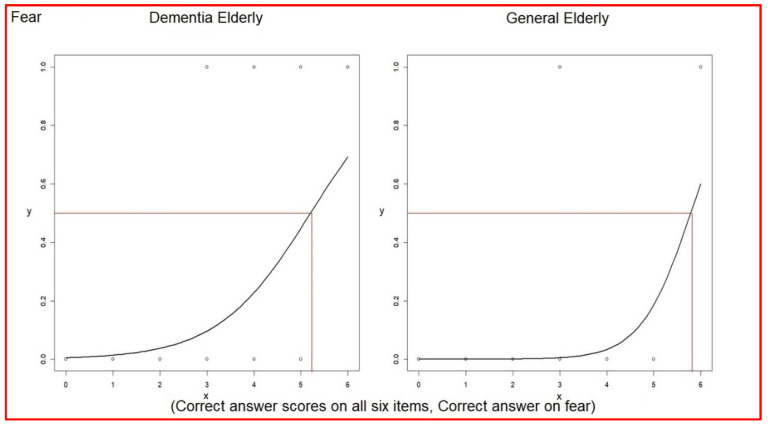
Logistic regressions of each correct answer versus correct answer scores on fear.

**Figure 2 healthcare-10-01022-f002:**
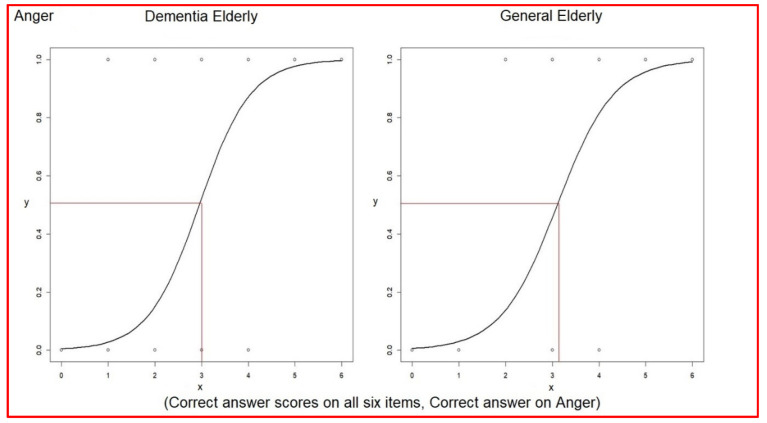
Logistic regressions of each correct answer versus correct answer scores on anger.

**Figure 3 healthcare-10-01022-f003:**
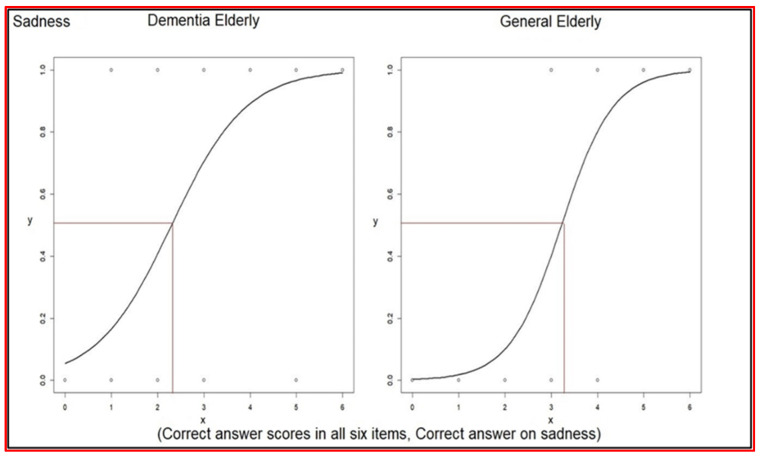
Logistic regressions of each correct answer versus correct answer scores on sadness.

**Figure 4 healthcare-10-01022-f004:**
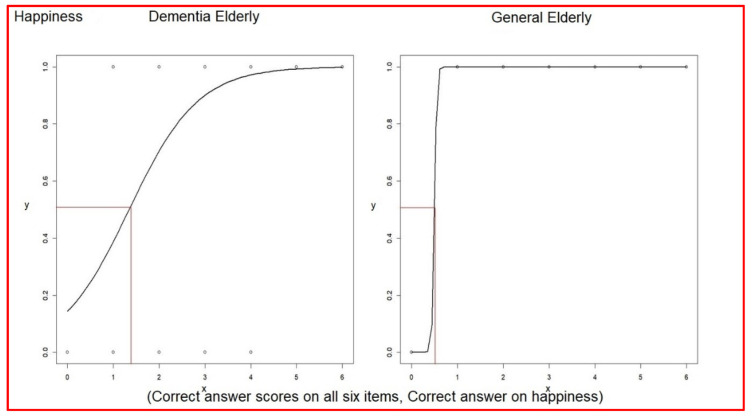
Logistic regressions of each correct answer versus correct answer scores on happiness.

**Figure 5 healthcare-10-01022-f005:**
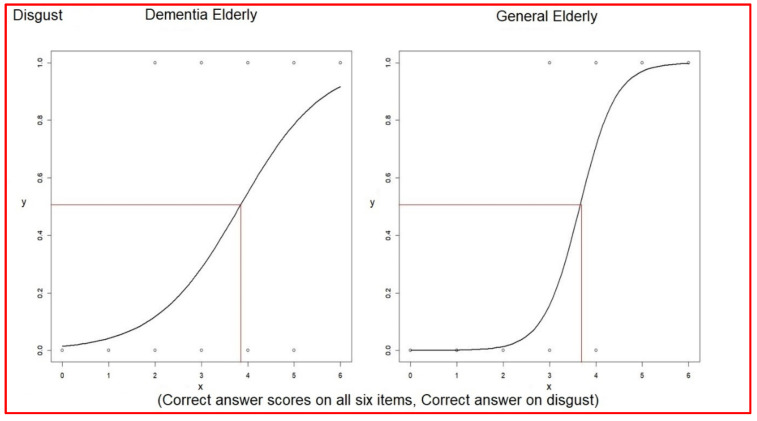
Logistic regressions of each correct answer versus correct answer scores on disgust.

**Figure 6 healthcare-10-01022-f006:**
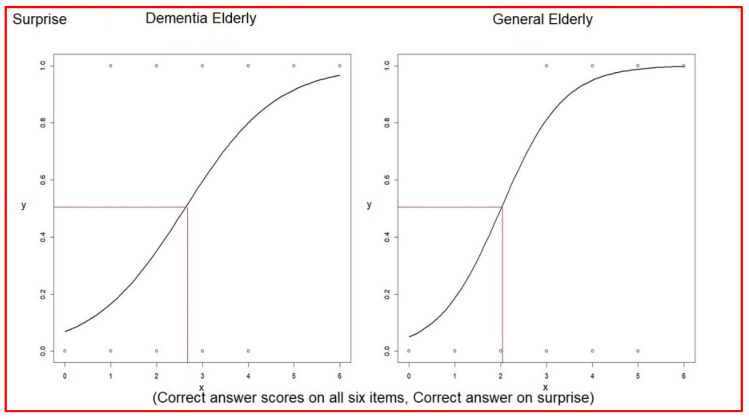
Logistic regressions of each correct answer versus correct answer scores on surprise.

**Table 1 healthcare-10-01022-t001:** Characteristics of subjects (*n* = 141).

Category	Subcategory	Elderly Dementia Patients (*n* = 72)	General Elderly Group (*n* = 69)	χ^2^	*p*
		*n*(%)	*n*(%)		
Sex	Female	45 (62.5)	52 (75.3)	2.14	0.142
	Male	27 (37.5)	17 (24.6)		
Age	70 s	31 (43.0)	41 (59.4)	3.14	0.075
	80 s	41 (56.9)	28 (40.5)		
Long term care grade	6	9 (12.5)	NA	NA	NA
	5	12 (16.6)			
	4	23 (31.9)			
	3	28 (38.8)			
Educational degree [39]	≤Elementary	26 (36.1)	22 (31.8)	6.46	0.010
	≥Middle	19 (26.3)	47 (68.1)		
	Missingvalues	27 (37.5)	0 (0.0)		
Income	<100	58 (80.5)	53 (76.8)	0.45	0.498
(thousand won)	≥100	12 (16.6)	16 (23.1)		
	Missing value	2 (2.7)	0 (0.0)		

**Table 2 healthcare-10-01022-t002:** Correct associations between verbal expressions to facial expressions (*n* = 141).

Expression Type	Elderly Dementia Patients (*n* = 72)	General Elderly Group (*n* = 69)	χ^2^	*p*
	% (*n*)	% (*n*)		
Fear	15.2 (11)	14.4 (10)	0.04	0.827
Anger	45.8 (33)	75.3 (52)	4.24	0.039
Sad	59.4 (41)	73.9 (51)	1.08	0.297
Happy	72.2 (52)	97.1 (67)	1.89	0.169
Disgust	31.9 (23)	66.6 (46)	7.66	0.005
Surprise	51.3 (37)	89.8 (62)	6.31	0.011

**Table 3 healthcare-10-01022-t003:** Correct associations between verbal expressions and facial expressions in elderly dementia patients (*n* = 72).

Verbal Expressions	Facial Expressions					
	Fear	Anger	Sad	Happy	Disgust	Surprise
	% (*n*)	% (*n*)	% (*n*)	% (*n*)	% (*n*)	% (*n*)
Fear	15.2 (11)	9.7 (7)	12.5 (9)	2.7 (2)	13.8 (10)	8.3 (6)
Anger	9.7 (7)	45.8 (33)	2.7 (2)	2.7 (2)	22.2 (16)	6.9 (5)
Sad	6.9 (5)	11.1 (8)	59.4 (41)	4.1 (3)	11.1 (8)	6.9 (5)
Happy	5.5 (4)	4.1 (3)	4.1 (3)	72.2 (52)	2.7 (2)	4.1 (3)
Disgust	12.5 (9)	11.1 (8)	8.3 (6)	8.3 (6)	31.9 (23)	5.5 (4)
Surprise	41.6 (30)	9.7 (7)	2.7 (2)	2.7 (2)	9.7 (7)	51.3 (37)
Not answered	5.5 (4)	5.5 (4)	9.7 (7)	4.1 (3)	5.5 (4)	13.8 (10)

**Table 4 healthcare-10-01022-t004:** Correct associations between verbal expressions and facial expressions in the general elderly group (*n* = 69).

Verbal Expressions	Facial Expressions					
	Fear	Anger	Sad	Happy	Disgust	Surprise
	% (*n*)	% (*n*)	% (*n*)	% (*n*)	% (*n*)	% (*n*)
Fear	14.4 (10)	5.7 (4)	4.3 (3)	0.0 (0)	2.8 (2)	2.8 (2)
Anger	0.0 (0)	75.3 (52)	4.3 (3)	0.0 (0)	15.9 (11)	2.8 (2)
Sad	8.6 (6)	7.2 (5)	73.9 (51)	2.8 (2)	8.6 (6)	0.0 (0)
Happy	0.0 (0)	2.8 (2)	1.4 (1)	97.1 (67)	4.3 (3)	2.8 (2)
Disgust	1.4 (1)	5.7 (4)	14.4 (10)	0.0 (0)	66.6 (46)	1.4 (1)
Surprise	75.3 (52)	2.8 (2)	1.4 (1)	0.0 (0)	1.4 (1)	89.8 (62)

**Table 5 healthcare-10-01022-t005:** Ratio of correct associations between verbal and facial expressions.

Verbal Expressions	Fear	Anger	Sad	Happy	Disgust	Surprise
	% (*n*)	% (*n*)	% (*n*)	% (*n*)	% (*n*)	% (*n*)
Fear	1.05					
Anger		0.60				
Sad			0.80			
Happy				0.74		
Disgust					0.47	
Surprise						0.57

**Table 6 healthcare-10-01022-t006:** Correct associations between verbal expressions to facial expressions across dementia grades (*n* = 72).

Severity	Dementia Grades	Verbal Expressions to Facial Expressions						
		Fear	Anger	Sad	Happy	Disgust	Surprise	M ± SD
		% (*n*)	% (*n*)	% (*n*)	% (*n*)	% (*n*)	% (*n*)	
Moderate	3 (*n* = 28)	17.8	50.0	60.7	64.2	28.5	42.8	44.0
		(5)	(14)	(17)	(18)	(8)	(12)	
	4 (*n* = 23)	17.3	43.4	52.1	78.2	39.1	73.9	50.7
		(4)	(10)	(12)	(18)	(9)	(17)	
Mild	5 (*n* = 12)	16.6	33.3	58.3	91.6	25.0	58.3	47.2
		(2)	(4)	(7)	(11)	(3)	(7)	
	6 (*n* = 9)	0.0	55.5	55.5	55.5	33.3	11.1	35.1
		(0)	(5)	(5)	(5)	(3)	(1)	

**Table 7 healthcare-10-01022-t007:** Correct associations between verbal expressions and facial expressions between positive and negative emotions (*n* = 141).

Emotion Type		Dementia (*n* = 72)		General (*n* = 69)		
		%	M ± SD	%	M ± SD	χ^2^ (*p*)
Positive	Happy	72.2	61.7	97.1	93.4	7.33 (0.006)
	Surprise	51.3		89.8		
Negative	Fear	15.2	38.0	14.4	57.5	9.74 (0.001)
	Anger	45.8		75.3		
	Sad	59.4		73.9		
	Disgust	31.9		66.6		

**Table 8 healthcare-10-01022-t008:** ROC for scores of correct associations between verbal expressions and facial expressions between elderly dementia patients and the general elderly group.

Cutoff	Sensitivity	Specificity	Positive Predictability	Negative Predictability	AUC
5.5	0.95	0.11	0.27	0.46	0.74
4.5	0.83	0.46	0.27	0.38	
3.5	0.68	0.72	0.31	0.27	
2.5	0.41	0.92	0.39	0.14	
1.5	0.29	0.94	0.43	0.16	
0.5	0.12	0.97	0.48	0.18	

AUC: Area under the curve.

**Table 9 healthcare-10-01022-t009:** Logistic regressions of each correct answer versus correct answer scores on all six items.

Division			Estimate	SE	z	*p*	EO	OR	(95% CI)
Dementia elderly patients									
Fear	Score	Intercept	−5.28	1.34	−3.91	<0.001	5.22	2.75	(1.57, 5.68)
		Degree	1.01	0.31	3.19	0.001			
Anger	Score	Intercept	−5.38	1.38	−3.89	<0.001	2.95	6.20	(3.00, 18.58)
		Degree	1.82	0.44	4.05	<0.001			
Sadness	Score	Intercept	−2.85	0.76	−3.74	<0.001	2.29	3.46	(2.13, 6.51)
		Degree	1.24	0.28	4.43	<0.001			
Happiness	Score	Intercept	−1.78	0.63	−2.80	0.005	1.34	3.77	(2.19, 7.68)
		Degree	1.32	0.31	4.22	<0.001			
Disgust	Score	Intercept	−4.23	0.99	−4.24	<0.001	3.84	3.02	(1.88, 5.65)
		Degree	1.10	0.27	4.00	<0.001			
Surprise	Score	Intercept	−2.61	0.69	−3.75	<0.001	2.63	2.71	(1.79, 4.58)
		Degree	0.99	0.23	4.25	<0.001			
General elderly group									
Fear	Score	Intercept	−10.85	3.21	−3.37	<0.001	5.80	6.53	(2.30, 26.55)
		Degree	1.87	0.61	3.05	0.002			
Anger	Score	Intercept	−5.13	1.70	−3.01	0.002	3.10	5.23	(2.41, 14.98)
		Degree	1.65	0.45	3.60	<0.001			
Sadness	Score	Intercept	−5.78	1.79	−3.21	0.001	3.22	6.02	(2.67, 18.13)
		Degree	1.79	0.48	3.71	<0.001			
Happiness	Score	Intercept	−20.22	14915	−0.001	0.999	0.49	NA	(NA, NA)
		Degree	40.70	19153	0.002	0.998			
Disgust	Score	Intercept	−9.40	2.44	−3.84	<0.001	3.65	13.14	(4.59, 58.44)
		Degree	2.57	0.63	4.07	<0.001			
Surprise	Score	Intercept	−2.95	1.63	−1.80	0.071	2.00	4.35	(1.96, 15.44)
		Degree	1.47	0.49	2.96	0.003			

SE: Standard error, EO: Even odd, OR: Odds ratio, CI: Confidence interval.

**Table 10 healthcare-10-01022-t010:** Scores between verbal expressions to facial expressions between dementia elderly patients and the general elderly group (*n* = 141).

Dementia Elderly Patients (*n* = 72) M ± SD	General Elderly Group (*n* = 69) M ± SD	t	*p*
2.69 ± 1.70	4.14 ± 1.34	5.63	<0.001

## Data Availability

Informed consent was obtained from all subjects involved in the study.

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
