# Peer review of "Study on Associating Emotions in Verbal Reactions to Facial Expressions in Dementia"

_healthcare, 2022, doi:10.3390/healthcare10061022_

Round 1

Reviewer 1 Report

The author compare the elderly with dementia verbal response to facial expression with controls. The study is valuable, and it encompass rich subjects, both in control and patients cohort. But I still have also some questions/suggestions for the paper.

  1. Dementia is 39 recognized as both a national problem as well as a problem family problem. In Introduction paragraph 1, is it some editing errors?
  2. The study subject is very close to one previous published work DOI: https://doi.org/10.1017/S1041610292001042

Facial Recognition: A Cognitive Study of Elderly Dementia Patients and Normal Older Adults.

Could the author please explain the advantage of the current study than before? The authors should cite these relevant works.

  1. The figures quality is too low. Especially the annotation in Fig1-6
  2. Is the table2. Percentage and (n) consistent? For example Sad 59.4%(11).

Author Response

Dear reviewer.

Point 1.

Dementia is 39 recognized as both a national problem as well as a problem family problem. In Introduction paragraph 1, is it some editing errors?

Response 1.

In 39-40, 1-1. Need for study, 1. Introduction, “Dementia is recognized as both a national problem as well as problem family problem.”. The above sentence is editing error. Hence, we correct this sentence as the below. “Dementia is recognized as a national problem as with a family problem.”.

Point 2.

The study subject is very close to one previous published work. DOI: https://doi.org/10.1017/S1041610292001042 Facial Recognition: A Cognitive Study of Elderly Dementia Patients and Normal Older Adults. Could the author please explain the advantage of the current study than before? The authors should cite these relevant works.

Response 2.

I inserted the below statement on discussion and reference on that Zandi et al.(1992)’ work. “In this study, subjects chose verbal expression on each of six face pictures. We identified the difference between dementia patients and control in the above condition. And furthermore, identified the face pictures causing the difference between two groups in the average scores on the even odds.”

References

  1. Zandi T, Cooper M, Garrison L(1992). Facial Recognition: A Cognitive Study of Elderly Dementia Patients and Normal Older Adults. International Psychogeriatrics. 4(2): 215-221.

Point 3.

The figures quality is too low. Especially the annotation in Fig1-6

Response 3.

I changed captions from small, bold to large, normal ones on figures. 

Point 4.

Is the table2. Percentage and (n) consistent? For example Sad 59.4%(11).

Response 4.

I am sorry for editing error. We corrected 59.4%(11) to 59.4%(41) of n=72.

Thank you.

Reviewer 2 Report

The study herein report show that elderly persons with dementia have difficulties identifying negative moud or expression face and identify and relate much easier with positive emotions, happiness represented by smilly faces. Those seems to be the conclusions of the study. However authors stress that "Further studies are needed regarding the use of two facial expressions (anger, disgust) as a simple screening test for the recognition of facial expression in dementia patients" and state that "As a result, our study showed that the ability of elderly dementia patients to recognize and associate emotions in verbal expressions with facial expressions is different from that of the general elderly population." which is a rather general idea. Although the work is sound, including metrologicaly, and that the subject is of particular difficulty, the work needs to be continued and evolve until clear and objective conclusion can be stated. I strongly advise authors to further extend their research and set more enphatically their conclusions.
Without giving a number for comparaison the statement (line 22) "The average score of elderly with dementia in associating verbal expressions with facial expressions was 2.69," have no practical interest in the abstract.

Author Response

Dear reviewer.

Point 1.

The study herein report show that elderly persons with dementia have difficulties identifying negative mood or expression face and identify and relate much easier with positive emotions, happiness represented by smiley faces. Those seems to be the conclusions of the study. However authors stress that "Further studies are needed regarding the use of two facial expressions (anger, disgust) as a simple screening test for the recognition of facial expression in dementia patients" and state that "As a result, our study showed that the ability of elderly dementia patients to recognize and associate emotions in verbal expressions with facial expressions is different from that of the general elderly population." which is a rather general idea. Although the work is sound, including metrologically, and that the subject is of particular difficulty, the work needs to be continued and evolve until clear and objective conclusion can be stated. I strongly advise authors to further extend their research and set more emphatically their conclusions.
Without giving a number for comparison the statement (line 22) "The average score of elderly with dementia in associating verbal expressions with facial expressions was 2.69," have no practical interest in the abstract.

Response 1.

We were very careful to sate conclusion. We appreciate your review and recommended conclusion. We decided to insert the next statement as conclusion as you recommended. “This study shows that elderly persons with dementia have difficulties identifying two negative emotions(anger, disgust) and have much easier identifying a positive emotion of happiness represented by smiling face.”. We will start this statement at the beginning of Conclusion and insert this statement in the abstract.

Thank you.

Reviewer 3 Report

Dear authors,

Congratulations for your paper. I wish to have more studies as regards the use of some facial expressions, as anger or disgust, however these will be something to manage in the future.

Some comments for improving the article are:

-  Some sentences may be improved and make them more cohesive.

- All the figures captions must be rewritten.

- All figures mention in the vertical axis facial expressions, however the way to determine them is not clear.

- The references are extensive, however they should written using the same APA format.

- Some tables, as table 7, need to reduce the text size. 

Author Response

Dear reviewer.

Point 1.

Some sentences may be improved and make them more cohesive.

Response 1.

1) In Abstract and Conclusion, We will start the next statement at the beginning of Conclusion and insert them in the abstract. “This study shows that elderly persons with dementia have difficulties identifying two negative emotions(anger, disgust) and have much easier identifying a positive emotion of happiness represented by smiling face.”. 2) In 39-40, 1-1. Need for study, 1. Introduction, “Dementia is recognized as both a national problem as well as problem family problem.”. The above sentence is editing error. Hence, we correct this sentence as the below. “Dementia is recognized as a national problem as with a family problem.”.

Point 2.

All the figures captions must be rewritten. All figures mention in the vertical axis facial expressions, however the way to determine them is not clear.

Response 2.

I changed captions from small, bold to large, normal ones on figures. And vertical axis is y, horizontal axis x. (x, y) is (Correct answer scores on all six items, Correct answer on happiness.

Point 3.

The references are extensive, however they should written using the same APA format.

Response 3.

We corrected references format. And We inserted DOI as possible: For examples:

  1. National Institute of Dementia(2019). Korean Dementia Observatory 2018. https://ansim.nid.or.kr/pds_view.aspx?page=&BID=194

  1. Carr AR, Ashla MM, Jimenez EE & Mendez MF(2018). Screening for Emotional Expression in Frontotemporal Dementia: A Pilot Study. Hindawi Behavioural Neurology. 2018 Mar 1;2018:8187457. https://doi.org/10.1155/2018/8187457

  1. Vocabular.com(n.d.). facial expression. In Vocabulary.com dictionary. Retrieved January 15 2019, from https://www.vocabulary.com/dictionary/facial expression.

Point 4.

Some tables, as table 7, need to reduce the text size. 

Response 4.

We reduced the height of table7.

Table 7. Correct associations between verbal expressions and facial expressions between positive and negative emotions (N=141).

Dementia(n=72)

General(n=69)

%

M±SD

%

M±SD

χ2(p)

Positive

Happy

72.2

61.7

97.1

93.4

7.33(.006)

Surprise

51.3

89.8

Negative

Fear

15.2

38.0

14.4

57.5

9.74(.001)

Anger

45.8

75.3

Sad

59.4

73.9

Disgust

31.9

66.6

Thank you.